# Transcriptomic Identification of Potential C2H2 Zinc Finger Protein Transcription Factors in *Pinus massoniana* in Response to Biotic and Abiotic Stresses

**DOI:** 10.3390/ijms25158361

**Published:** 2024-07-31

**Authors:** Dengbao Wang, Zimo Qiu, Tao Xu, Sheng Yao, Meijing Chen, Qianzi Li, Romaric Hippolyte Agassin, Kongshu Ji

**Affiliations:** 1State Key Laboratory of Tree Genetics and Breeding, Nanjing Forestry University, Nanjing 210037, China; dbw@njfu.edu.cn (D.W.); zmqiu@njfu.edu.cn (Z.Q.); 2110103121@njfu.edu.cn (T.X.); yaosheng0817@163.com (S.Y.); cmj0302@njfu.edu.cn (M.C.); qianzili@njfu.edu.cn (Q.L.); hippolyteagassin@gmail.com (R.H.A.); 2Key Open Laboratory of Forest Genetics and Gene Engineering of National Forestry & Grassland Administration, Nanjing 210037, China; 3Co-Innovation Center for Sustainable Forestry in Southern China, Nanjing Forestry University, Nanjing 210037, China

**Keywords:** C2H2 zinc finger proteins, expression patterns, *Pinus massoniana*, abiotic stress

## Abstract

Biotic and abiotic stresses have already seriously restricted the growth and development of *Pinus massoniana*, thereby influencing the quality and yield of its wood and turpentine. Recent studies have shown that C2H2 zinc finger protein transcription factors play an important role in biotic and abiotic stress response. However, the members and expression patterns of C2H2 TFs in response to stresses in *P. massoniana* have not been performed. In this paper, 57 C2H2 zinc finger proteins of *P. massoniana* were identified and divided into five subgroups according to a phylogenetic analysis. In addition, six Q-type *PmC2H2-ZFPs* containing the plant-specific motif ‘QALGGH’ were selected for further study under different stresses. The findings demonstrated that *PmC2H2-ZFPs* exhibit responsiveness towards various abiotic stresses, including drought, NaCl, ABA, PEG, H_2_O_2_, etc., as well as biotic stress caused by the pine wood nematode. In addition, *PmC2H2-4* and *PmC2H2-20* were nuclear localization proteins, and *PmC2H2-20* was a transcriptional activator. *PmC2H2-20* was selected as a potential transcriptional regulator in response to various stresses in *P. massoniana*. These findings laid a foundation for further study on the role of *PmC2H2-ZFPs* in stress tolerance.

## 1. Introduction

*Pinus massoniana*, a widely distributed coniferous tree in southern China, is not only an economically important species for timber, wood pulp, and rosin but also plays a significant ecological role in the forest ecosystem [1]. The occurrence of seasonal soil drought in Southern China poses a significant natural constraint on the production and growth of Masson pine [2]. In addition, the prevalence of pine wood disease in China over the past four decades has caused a tremendous disaster for coniferous plants. Therefore, it is highly necessary to cultivate drought-resistant and disease-resistant Masson pine. A large number of transcription factors (TFs) were involved in the response to biotic and abiotic stresses [3,4,5,6,7,8,9,10,11,12,13]. Recent studies regarding comprehensive genome-wide sequencing, various transcriptome analyses, and the functional identification of numerous genes have demonstrated the involvement of various transcription-factor families in conferring abiotic stress tolerance in plants, including MYB [4,5], AP2/ERF [7], NAC [8], bHLH (basic helix–loop–helix) [10], bZIP (basic leucine zipper) [11], homeodomain [13], WRKY [14], HSF [15], MADS-box [16], and zinc finger proteins [17]. The large and diverse zinc finger protein family plays important roles in all aspects of plant growth and development [18]. The first zinc finger protein (TF IIIA) was initially observed in *Xenopus laevis* oocytes in 1985, and the term ‘zinc finger’ was coined to describe their distinctive structural resemblance to fingers [19]. Since then, zinc fingers with a variety of functions have been found in animals, plants, yeasts, and viruses [20].

Zinc finger proteins (ZFPs) play a critical role in transcriptional regulation, RNA binding, the regulation of apoptosis, and protein–protein interactions [21]. They can be classified as C2H2, C2C2, C2HC, C8, C2HC5, C3HC4, CCCH, C6, C4, and C4HC3 [22,23]. Among these types, C2H2-type zinc finger proteins are widely recognized as one of the most abundant and extensively investigated protein families in eukaryotes [8]. C2H2 ZFPs contain two cysteine (C) and two histidine (H) residues in each finger, and these residues bind a zinc ion that stabilizes the ZFP and specifically binds to a domain within the promoter of the gene it regulates [24]. In plants, numerous C2H2-ZFPs exhibit conserved structural features among themselves. Its sequence is characterized by C-X_2-4_-C-X_12_-H-X_3-5_-H (X represents any amino acid) [25]. In multiple-fingered proteins, the adjacent fingers are separated by a long spacer that is highly variable in length and sequence from each other [21]. Most fingers have a six-amino acid stretch, ‘QALGGH’, at a position corresponding to the N-terminal part of the recognition helix. They are called Q-type C2H2 ZFPs [26].

Currently, an increasing number of Q-type C2H2 ZFPs have been identified in various plants. *Arabidopsis thaliana* has been reported to possess a total of 176 C2H2 ZFPs [27], while 189, 109, 321, 54, and 47 C2H2 ZFPs have been identified in rice (*Oryza sativa*) [28], poplar (*Populus trichocarpa*) [29], soybean (*Glycine max*) [30], apple (*Malus domestica* Borkh.) [31] and wheat (*Triticum aestivum*) [32], respectively. Recent functional analysis has shown that C2H2-ZFPs are involved in regulating multiple growth-development processes and resisting biotic and abiotic stress in plants [12,20]. *EPF1* was identified from *Petunia* as the first plant-specific ZFP that interacted with the promoter region [33]. Later, WZF1 was reported in wheat as a DNA-binding zinc finger protein that interacts with a cis element of histone genes [34]. *C2H2-ZFP245* might function as a downstream component of CBF/DREB proteins by repressing the expression of some genes in response to cold or drought stress in *O. sativa* [35]. The overexpression of *C2H2-SlZF3* in *Solanum lycopersicum* and *A. thaliana* resulted in the accumulation of AsA, thereby alleviating the oxidative damage caused by salt stress through the enhancement of the transgenic plants’ ROS-scavenging capacity [36]. In *Glycine max*, *C2H2-GmSCOF-1* was induced by low temperature and abscisic acid (ABA) treatments but not by dehydration or high salinity [37]. These results suggest that C2H2-ZFPs act as transcriptional activators or repressors in different stress signal-transduction pathways to regulate the transcriptional levels of downstream genes.

However, the identification of C2H2-ZFPs and stress-related functional analysis in *P. massoniana* is still lacking. In recent years, the advancements in transcriptome and genome analysis have provided us with an opportunity to identify TF families in plants that play important roles in diverse stress responses. In this study, 57 *PmC2H2-ZFPs* were identified from three stress-related transcriptomes and characterized using bioinformatic analysis. Then, the expression profiles of Q-type *PmC2H2* in different tissues and stresses were studied to better understand their roles in regulating *P. massoniana* stress response. The results of this study provide insights into the characterization of *PmC2H2-ZFPs*, as well as screening a potential regulatory gene in response to biotic and abiotic stresses in *P. massoniana*.

## 2. Results

### 2.1. Transcriptome-Wide Identification and Analysis of C2H2 ZFPs in P. massoniana

After excluding incomplete and highly homologous sequences (>97%), further confirmation of the conserved domain was obtained through prediction from SMART and CD-search. We identified 57 *C2H2 ZFPs* with a highly conserved C2H2 domain (e-value < 0.001) from three transcriptomes in response to biotic and abiotic stresses in *P. massoniana* (Appendix A). We designated them as *PmC2H2*-1 to *PmC2H2*-57 (Table 1). These C2H2 ZFPs exhibited significant variations in length, ranging from 239 to 725 amino acids (Aa), with an average length of 477.34 aa. The molecular weight (MW) of *PmC2H2s* ranged from 27.41 kDa to 81.74 kDa, while the isoelectric point (pI) values ranged from 4.82 to 9.46. Subsequently, subcellular localization prediction results revealed that the majority of *PmC2H2s* were localized in the nucleus, except for one instance where *PmC2H2-29* was found to be localized in the chloroplast.

### 2.2. Phylogenetic Analysis and Domain Analysis of PmC2H2s

A phylogenetic tree was constructed using the ML method, as shown in the left part of Figure 1, which facilitated the classification of these 57 PmC2H2 ZFPs into five distinct subgroups denoted as ‘I’ to ‘V’. Among them, group I represents the largest subgroup, comprising a total of 33 C2H2-ZFP members. The smallest group was Group III, which consisted of *PmC2H2-8* and *PmC2H2-43*. Groups II, IV, and V contain 11, 8, and 3 members, respectively. According to the MEME program, for the identification of the conserved motifs of 57 C2H2 ZFPs (Figure 1), the amino acid length of the 10 motifs ranged from 8 to 50 (Table 2). We found that all members had at least one highly typical conserved C2H2 domain (motif 1). Motif 2 was distributed in all groups. In addition, motifs 5 and 7 were only found in group I. Motif 3 and motif 6 were found in six members in group III with an exception in group I (PmC2H2-12). In addition, we found that six members (PmC2H2-4, PmC2H2-5, PmC2H2-16, PmC2H2-20, PmC2H2-24, and PmC2H2-33) contained the ‘QALGGH’ domain in group IV; they are also called Q-type C2H2 ZFPs. The occurrence of motif 4 was frequently observed in groups I and II. Motif 8 was distributed in all groups except Group V. Motif 9 occurs primarily in group I, with an exception in group II (PmC2H2-52). Motif 10 was distributed in group II and group V. In order to further determine the type and distribution of the C2H2 domains, a regular expression search showed that the most common C2H2 domain is C-X_2_-C-X_12_-H-X_3_-H, accounting for 49% of all C2H2, followed by C-X4-C-X12-H-X3-H and C-X4-C-X12-H-X4-H, accounting for 19% and 14%, respectively. However, the structure of C-X3-C-X12-H-X3-H and C-X3-C-X12-H-X5-H have not been found in Masson pine (Appendix A). Considering that Q-type C2H2 is unique to plants and is widely involved in plant responses to both biotic and abiotic stresses, the functional studies of zinc finger proteins have predominantly focused on Q-type C2H2 members. However, there are no studies of Q-type C2H2 ZFPs that have been reported on Masson pine yet. Therefore, we selected these six Q-type members for further study (Appendix A).

### 2.3. Subcellular Localization Analysis

Subcellular localization prediction indicated that most C2H2 TFs likely exert regulatory functions at the nucleus. In order to further comprehend the localization features of PmC2H2 ZFPs, *PmC2H2-4* and *PmC2H2-20* were selected for the experiment. A fluorescent signal was found after transient transformation in tobacco leaves. The GFP signal was distributed throughout the whole cell in the control; however, mGFP5 fused with *PmC2H2-4* and *PmC2H2-20* only showed fluorescence in the nucleus (Figure 2). The results indicated that *PmC2H2-4* and *PmC2H2-20* are nuclear localization proteins.

### 2.4. Expression Patterns of PmC2H2 ZFPs under Nematode and Drought Stress

Drought and nematodes are the two most important factors currently limiting the growth and development of the Masson pine. Therefore, two heatmaps were generated to analyze the expression levels of *PmC2H2 ZFPs* under nematode infestation and drought stress conditions, based on the transcriptome data (Figure 3). The transcriptome data represent gene-expression patterns in a specific temporal and spatial context, so only 34 and 22 expression patterns of *PmC2H2 ZFPs* were obtained from the RNA-seq transcriptomics data, respectively. After infection by nematodes, 18 members of the C2H2 family showed induced expression at different time points. The expression of 5 members was suppressed after nematode infection, while the expression levels of 11 members initially increased and then decreased over time. As the drought intensifies, the expression levels of 15 members tend to increase, while the expression levels of 7 members decrease.

### 2.5. Expression Patterns of Q-Type C2H2 ZFPs Genes in Different Tissues

The qRT-PCR analysis in Figure 4 showed the expression patterns of six Q-type *PmC2H2s* in eight different tissues: shoot apices (T); young needle leaves (YL); old needle leaves (OL); young stem (YS); old stem (OS); xylem (X); phloem (P); and root (R). It is evident from the results that *PmC2H2-4*, *PmC2H2-5*, and *PmC2H2-20* exhibit high expression in needles. *PmC2H2-16* and *PmC2H2-20* expressed significantly in the root, while *PmC2H2-33* expressed highly in the young stem. The expression levels of *PmC2H2-4* and *PmC2H2-5* in the phloem are extremely low, while *PmC2H2-20* and *PmC2H2-24* have the lowest expression in the xylem. In addition, *PmC2H2-33* exhibited a low expression in the old stem. Notably, the expression of *PmC2H2-16* in stems and the xylem was almost undetectable.

### 2.6. Expression Levels of Q-Type C2H2 ZFPs Genes under Abiotic Stresses

Figure 5 shows the expression patterns of six genes under different abiotic stresses. *PmC2H2-20* and *PmC2H2-24* were induced significantly by ABA treatment (Figure 5a), while other genes exhibited lower expression levels under ABA treatment. Under natural drought conditions, with the exception of *PmC2H2-20* which showed suppressed expression on the 20th day of drought, other genes were induced at different stages of drought. *PmC2H2-20* and *PmC2H2-33* were suppressed significantly by ETH (Figure 5c), and *PmC2H2-5* was not sensitive to ETH treatment. The expressions of *PmC2H2-4*, *PmC2H2-16,* and *PmC2H2-24* increased significantly at first and then decreased after 6 h. The expression of *PmC2H2-5* and *PmC2H2-16* were suppressed under H_2_O_2_ treatment, while other genes were induced by H_2_O_2_ first, and then, their expression presented a downward trend (Figure 5d). The *PmC2H2-20* gene exhibited no sensitivity to mechanical injury stress (Figure 5e), while the expression levels of the remaining five genes displayed an initial increase followed by a subsequent decrease over time, with peak expression observed at either 3 h or 6 h. The expressions of *PmC2H2-16*, *PmC2H2-20*, and *PmC2H2-33* were inhibited by MeJA (Figure 5f), and the expression levels of *PmC2H2-24* increased significantly after MeJA treatment. The expression of *PmC2H2-20* was significantly inhibited under salt stress (Figure 5g), and the expression changes of *PmC2H2-5* under salt stress were not obvious. The expression levels of *PmC2H2-20* and *PmC2H2-33* significantly decreased after PEG treatment, while the other four genes were induced to varying degrees (Figure 5h). After SA treatment, it was worth noting that the expression level of *PmC2H2-20* significantly increased within 3 h and then recovered. The expression levels of *PmC2H2-20* were found to be significantly suppressed in most treatments, except for ABA and SA treatment, suggesting that *PmC2H2-20* may play a significant role in stress resistance in Masson pine.

### 2.7. Transcriptional Activity Analysis

The transcriptional activity of Q-type C2H2 was analyzed. We successfully cloned *PmC2H2-4*, *PmC2H2-5*, *PmC2H2-16*, *PmC2H2-20*, *PmC2H2-24*, and *PmC2H2-33* and constructed them into the PGBKT7 vector. The yeast strains that contained recombinant plasmids were cultured on the SD/-Trp medium first. Verified positive yeast colonies were collected onto the higher selective culture medium SD/-Trp/-His and SD/-Trp/-His/-Ala. The photographs of the yeast growth assay (Figure 6) showed that only *PmC2H2-20* can grow on the severe selective medium (SD/-Trp/-His/-Ala) and activate the expression of reporter genes, resulting in the production of β-galactosidase, which turns substrate x-α-gal blue, whereas the remaining five genes did not exhibit any self-activation. This result suggested that the *PmC2H2-20* can act as a transcriptional activator by binding to the promoter sequence of downstream reporter genes, thereby activating their transcription and helping *P. massoniana* resist stress.

## 3. Discussion

The C2H2-type zinc finger proteins are widely distributed among eukaryotes and play significant roles in diverse biological processes, including hormone signaling, DNA or RNA binding, and stress response [22,38,39,40]. Moreover, extensive exploration has been conducted to identify C2H2-type zinc finger proteins in various plant species. However, the lack of a reference genome for *P. massoniana* has created a knowledge gap in the comprehensive identification of C2H2 TF families in *P. massoniana* at the genomic level. Therefore, we can only temporarily fill this gap by using transcriptome data to identify stress-response TFs, thereby establishing a foundation for future research.

A total of 57 *PmC2H2 ZFPs* were identified from three transcriptomes and divided into five subgroups. Almost all *PmC2H2 ZFPs* were predicted to localize in the nucleus, supporting their role as a transcription factor in the nucleus. In addition, we confirmed the subcellular localization of *PmC2H2-4* and *PmC2H2-20* in tobacco leaves through instantaneous transformation, which is consistent with the prediction. In yeast and *A. thaliana*, C2H2 ZFPs were classified based on the quantity and configuration of ZF domains [41]. The C2H2 sequence is especially characterized by C-X_2,4_-C-X_12_-H-X_4-5_-H in *P. massoniana*. However, it lacks the forms of C-X_3_-C-X_12_-H-X_3_-H and C-X_3_-C-X_12_-H-X_5_-H compared to other plants. The majority type of conserved C2H2 domains exhibited in *P. massoniana* are C-X_2_-C-X_12_-H-X_3_-H.

In addition, previous researchers have demonstrated that the surrounding residues at the C-terminus of the ‘QALGGH’ motif play a pivotal role in facilitating specific DNA recognition [42]. The AGC(T) sequence was identified as the optimal binding site for the first zinc finger (ZF) of the Petunia ZPT2-2 protein, while the CAGT core was determined to be the preferred binding site for the second ZF [43]. This discovery confirms that flanking residues play a pivotal role in determining specificity for recognizing DNA target sequences and facilitating effective DNA binding [44]. We also identified six Q-type C2H2 ZFPs from Masson pine, all of which possess a ‘QALGGH’ motif. Current research has shown that Q-type C2H2 ZFPs are widely involved in plant biotic and abiotic stress responses. In addition, the expression of C2H2 ZFP is modulated by tissue-specific variations and abiotic stresses [45]. Consequently, we further investigated the expression patterns of six Q-type C2H2s under diverse stresses and tissues.

The role of *C2H2 ZFPs* as transcription factors in response to abiotic stresses has been extensively studied across various plant species [46]. The upregulation of a series of resistance genes can be induced by ABA, leading to their enhanced expression under conditions of drought and salt stress. This mechanism enhances the plant’s capacity to endure osmotic stress [47,48]. *STZ/ZAT10* derived from *A. thaliana* can serve as a supplementary factor for yeast calcineurin mutants, enhancing salt tolerance in transgenic yeast [21,49,50]. The expression of *LkZFP6* in *Larix kaempferi* can be induced by ABA treatment [51]. The qRT-PCR results revealed that *PmC2H2-20* and *PmC2H2-24* can be induced by ABA treatment, indicating that they may regulate Masson pine in response to abiotic stresses through the ABA-mediated signal pathway.

It is worth noting that *PmC2H2-20* exhibits significant inhibition under nematode stress and treatments other than ABA and SA, indicating its involvement in the biotic and abiotic stress response, such as drought, H_2_O_2_, damage, MeJA, NaCl, and PEG. In addition, *PmC2H2-24* can be induced by all treatments and nematode infection, suggesting its potential involvement in response to external stresses in *P. massoniana*. The stress-induced response pattern of the *PmC2H2-4* closely resembles that of *PmC2H2-24*, with the exception that its expression level is repressed upon ABA treatment. The remaining three Q-type C2H2 ZFPs exhibited distinct expression patterns in response to multiple stresses, implying the involvement of Q-type C2H2 genes in response to abiotic stresses.

The tissue-specific expression of genes usually is preliminarily used to predict their corresponding functions [52]. Previous research reported that the expression profiles of plant C2H2-ZFPs vary in different tissues and abiotic stresses [12,45]. A tissue-specific expression analysis showed that *PmC2H2-4*, *PmC2H2-5,* and *PmC2H2-20* may perform specific functions in needles. Whereas *PmC2H2-16* and *PmC2H2-24* function in the root, *PmC2H2-33* functions in the young stem. A previous report showed that *ZAT18* (AT3G53600) played a positive role in drought tolerance in *Arabidopsis* [53]. *PmC2H2-16* and *PmC2H2-24,* which are induced by drought, may be correlated with root growth and drought tolerance in *P. massoniana*.

In this study, we found the transcriptional activation of *PmC2H2-20* in the yeast system, indicating that *PmC2H2-20* could activate the expression of downstream reporter genes. The C2H2-type zinc protein of *A. thaliana* has been demonstrated in relevant studies to serve as a pivotal regulator of ROS signaling, thereby functioning as an integrative factor for coordinating responses to diverse abiotic stresses [54].

Although the role of *C2H2 ZFPs* has been extensively studied in model plants, such as *Arabidopsis* and *Oryza sativa*, there is little research on their involvement in pinaceae’s tolerance to both biotic and abiotic stresses. Our study has successfully identified 57 *C2H2 ZFPs* from the transcriptomes of Masson pine for the first time and analyzed their expression patterns under two major stress conditions currently restricting the development of *P. massoniana*. Furthermore, we have thoroughly investigated the expression patterns of Q-type C2H2 members under various abiotic stresses and screened a potential candidate gene, *PmC2H2-20*, which may play a crucial role as an important regulatory factor in the stress response of *P. massoniana*. These findings provide insights for future functional studies on the C2H2 genes involved in stress resistance mechanisms in *P. massoniana*.

## 4. Materials and Methods

### 4.1. Identification of C2H2 Genes in P. massoniana

We obtained the Hidden Markov Model (HMM) profile of the C2H2 domain (PF00096) from the Pfam database (http://pfam.xfam.org/) (accessed on 2 August 2023). The HMM profile was used to search for C2H2 proteins from three *P. massoniana* transcriptomes, namely CO_2_ stress transcriptome (SRA accession: PRJNA561037) [55], drought stress transcriptome (SRA accession: PRJNA595650) [56], and *P. massoniana,* inoculated with the pine wood nematode transcriptome (SRA accession: PRJNA660087). A BLASTP search was performed against four transcriptomes using the Hidden Markov Model (HMM) profile. We selected sequences with the C2H2 domain (e-value < 0.001) and deleted repeated sequences with a similarity of more than 97%. Transcription Factor Prediction (http://planttfdb.gao-lab.org/prediction.php) (accessed on 2 August 2023) was used to predict putative C2H2 proteins. Then, we use SMART (https://smart.embl.de/) (accessed on 2 August 2023) and NCBI Conserved Domain Search (CD Search) (https://www.ncbi.nlm.nih.gov/Structure/cdd/wrpsb.cgi) (accessed on 2 August 2023) to check the conserved domain of PmC2H2 TFs. Molecular weights and isoelectric points (pI) of identified PmC2H2 proteins were calculated using the ExPaSy site6 (https://web.expasy.org/compute_pi/) (accessed on 2 August 2023).

### 4.2. Phylogenetic and Bioinformatics Analysis

The maximum likelihood (ML) method implemented in MEGA-X was employed to sample 1000 bootstraps from the phylogenetic tree [57]. For visualization purposes, the phylogenetic tree underwent editing using EvolView online software (v2) (https://www.evolgenius.info/evolview) (accessed on 3 August 2023). To identify potential conserved motifs, we utilized the Multiple Expectation Maximization for Motif Elicitation (MEME) program (https://meme-suite.org/meme/tools/meme) (accessed on 3 August 2023) and the number of motifs was set to 10. Subcellular localization prediction and analysis of PmC2H2 protein were conducted using CELLO (http://cello.life.nctu.edu.tw/) (accessed on 3 August 2023), WoLF PSORT (https://wolfpsort.hgc.jp/) (accessed on 3 August 2023), and Plant-mPLoc tools (http://www.csbio.sjtu.edu.cn/bioinf/plant-multi/) (accessed on 3 August 2023).

### 4.3. Subcellular Localization Analysis

The open reading frame (ORF) regions of *PmC2H2-4* and *PmC2H2-20* without a stop codon were linked with the pCAMBIA-1302-mGFP5 vector (primers were listed in Appendix A). After PCR verification, the positive *Agrobacterium* was transferred to an LB medium supplemented with kanamycin (50 mg/L) and rifampicin (25 mg/L), followed by incubation until the optical density (OD) value reached 0.6. Subsequently, it was co-cultivated with the P19 (RNA Silencing Inhibitor) *Agrobacterium* strain in a suspension containing 150 µM acetosyringone, 10 mM MgCl2, and 10 mM 2-(N-morpholino) ethanesulfonic acid (MES). The leaves of 4-week-old *Nicotiana benthamiana* were injected with the mixed solution. Afterwards, the infiltrated *N. benthamiana* plants were kept in darkness for 48 h. The LSM710 confocal microscope (Zeiss, Jena, Germany) was used to capture the GFP signal.

### 4.4. RNA-Seq Data Analysis

The RNA-seq data of Masson pine under drought stress (PRJNA595650) were collected at four different field capacities representing natural drought conditions: CK (normal water supply) (80 ± 5)%, LD (light drought stress mild) (65 ± 5)%, MD (moderate drought stress) (50 ± 5)%, and SD (severe drought stress) (35 ± 5)%. In the case of pine wood nematode treatment (PRJNA660087), needles were sampled from each lateral branch at various time points (0 d, 3 d, 10 d, 20 d, and 35 d) after inoculation. Fragments per kilobase of the exon model per million reads mapped (FPKM) values were calculated to estimate the abundance of C2H2 ZFP transcripts. Heat maps depicting partial gene expression patterns based on log2(FPKM + 1) values were generated using TBtools software (v2.034) [58], with analyses performed at the row scale.

### 4.5. Plant Materials and Abiotic Stress Treatments

Two-year-old, healthy, and uniformly sized seedlings of *P. massoniana* were selected for tissue-specific analysis and abiotic treatments. The drought treatment was performed as follows. Needles were collected five times over a period of 20 days, allowing for natural evaporation after watering at day 0. The samples were collected at five time points with the field capacity as follows: 0 d (67%); 3 d (63%); 7 d (58%); 12 d (46%); and 20 d (34%). The osmotic stress was induced by soaking the seedlings in the 15% polyethylene glycol (PEG6000) solution and 200 mM NaCl solution. The mechanical damage treatment method was performed by cutting the upper half of the needles. For plant hormones treatment, the selected seedlings were sprayed independently with 100 µM ABA (abscisic acid); 1 mM SA (salicylic acid); 10 mM H_2_O_2_ (hydrogen peroxide); 10 mM MeJA (methyl jasmonate); and 50 µM ETH (ethephon) solutions (50 mL) on the surface of needles. Afterward, needles were sampled at 0 h, 3 h, 6 h, 12 h, and 24 h after treatment. All treatments were conducted on three biological replicates.

### 4.6. RNA Extraction and qRT-PCR Analysis

The total RNA of *P. massoniana* was extracted following the protocol provided in the FastPure plant total RNA isolation kit (RC401, Vazyme Biotech, Nanjing, China). The RNA concentration and purity were measured with a NanoDrop 2000 (Thermo Fisher Scientific, Waltham, MA, USA), and the RNA integrity was estimated by 1% agarose gel electrophoresis. First-strand cDNA was synthesized using the One-step gDNA Removal and cDNA Synthesis Kit (AT311, TransGen Biotech, Beijing, China). Primers for quantitative real-time reverse transcription PCR (qRT-PCR) were designed using Primer 5.0 (Appendix A). SYBR Green reagents were used to detect the target sequence. Each PCR mixture (10 µL) contained 1 µL of diluted cDNA (20× dilution), 5 µL of SYBR Green Master Mix (11184ES03, Yeasen Biotech, Shanghai, China), 0.4 µL of each primer (10 µM), and 3.2 µL of ddH_2_O. The PCR program stages were (1) 95 °C for 2 min (preincubation); (2) 95 °C for 10 s, and (3) 60 °C for 30 s, repeated 40 times. The remaining steps use the instrument’s default settings. The PCR quality was estimated based on the melting curves. The *alpha-tubulin* (TUA) gene was used as a reference gene [59]. Three independent biological replicates and three technical replicates for each biological replicate were examined. Quantification was achieved using comparative cycle threshold (Ct) values, and gene-expression levels were calculated as 2(−∆∆Ct) [∆CT = CT Target − CT TUA. ∆∆Ct = ∆Ct Target − ∆Ct CK]. Duncan’s test was used to examine the significance between different columns in IBM SPSS Statistics (Version 25). The lowercase letters represent the significance between different columns (*p* < 0.05).

### 4.7. Transcriptional-Activation Activity Assay

The open reading frames of *PmC2H2-4*, *PmC2H2-5*, *PmC2H2-16*, *PmC2H2-20*, *PmC2H2-24*, and *PmC2H2-33* were fused with the pGBKT7 vector (primer sequences were listed in Appendix A). Subsequently, these recombinant plasmids were transformed into AH109 yeast strains (YC1010, Weidi Biotech, Shanghai, China), which were cultured on the SD/-Trp medium at 29 °C for 48 h. Verified positive yeast colonies were collected into 200ul ddH_2_O. Subsequently, a repetitive procedure was performed to withdraw 5 μL from the suspension and deposit it onto the culture medium SD/-Trp/-His/-Ade, which contained X-α-Gal as an indicator. The empty pGBKT7 was used as a negative control, and *PmC3H20*, previously confirmed to exhibit transcriptional self-activation, was employed as a positive control [17].

## 5. Conclusions

In this study, we identified a total of 57 C2H2 ZFPs in *P. massoniana* and performed a bioinformatics analysis. These members were classified into five subgroups and six Q-type *C2H2* ZFPs were selected for further investigation to explore their expression patterns under various stress conditions and in different tissues. Subcellular localization prediction results revealed nuclear localization for all genes except *PmC2H2-29*, while experimental validation confirmed the nuclear localization of *PmC2H2-4* and *PmC2H2-20*. Two heatmaps showed that *PmC2H2s* could respond to pine wood nematodes stress and drought, respectively. However, we noticed that the expression levels of *PmC2H2-20* are down-regulated, except for its induction under ABA stress. Furthermore, the transcriptional activation activity of *PmC2H2-20* was detected, indicating its role as a potential transcriptional activator involved in stress-response mechanisms. This study establishes a theoretical basis for further investigation into C2H2 TFs associated with pine wood nematode and drought resistance in *P. massoniana*, while also presenting *PmC2H2-20* as a potential candidate gene that exhibits significant responsiveness to stress.

## Figures and Tables

**Figure 1 ijms-25-08361-f001:**
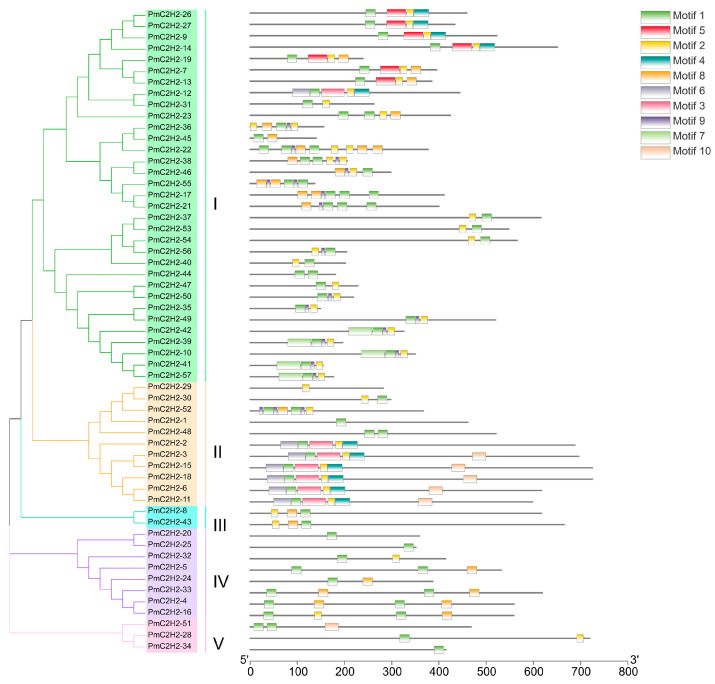
Phylogenetic analysis and motif distribution of PmC2H2 ZFPs. Branches with different colors represented different subgroups. The subgroup was named from “I” to “V”.

**Figure 2 ijms-25-08361-f002:**
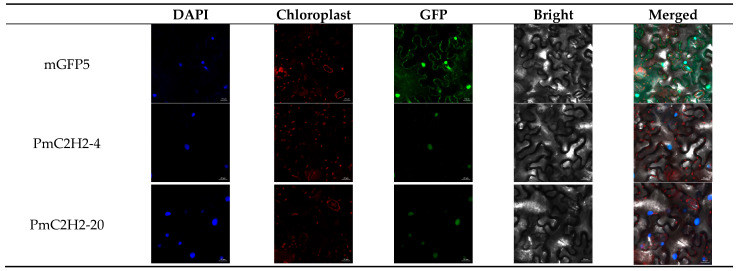
Subcellular localization analysis of *PmC2H2-4* and *PmC2H2-20* proteins in *N. benthamiana* leaves. The scale in the images is 20 μm. pCAMBIA-1302-mGFP5 was the control. DAPI—4′,6-diamidino-2-phenylindole, a blue fluorescent dye that shows DNA location. Chloroplast—chloroplast auto-fluorescence, displays the location of chloroplasts. GFP—green fluorescence protein, displays the location of the target protein. Bright—bright field. Merged—merged picture of four overlapped channels.

**Figure 3 ijms-25-08361-f003:**
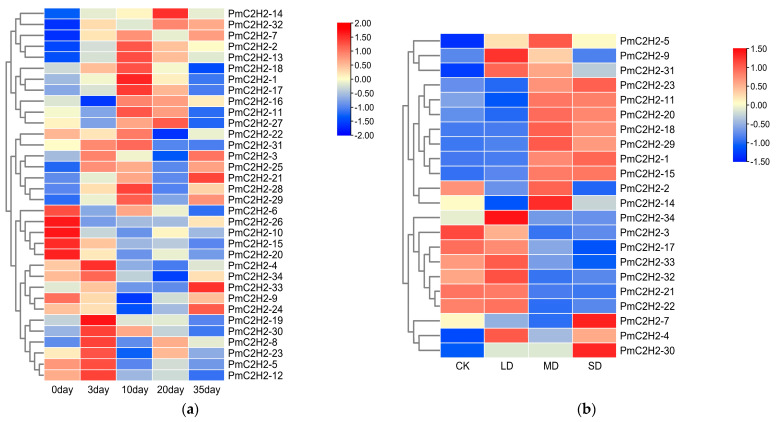
Transcriptional profiles of *C2H2 ZFPs* in *P. massoniana* under nematode stress and drought stress. (**a**) Different days after inoculation with pine wood nematodes: 0 (CK), 3, 10, 20, and 35 d. (**b**) Different soil water content represents different drought levels: CK (normal water supply) (80 ± 5)%, LD (light drought stress mild) (65 ± 5)%, MD (moderate drought stress) (50 ± 5)%, and SD (severe drought stress) (35 ± 5)%. A heatmap was generated using log2 (FPKM + 1) values, then normalized by row scale. The color scale represents relative expression levels.

**Figure 4 ijms-25-08361-f004:**
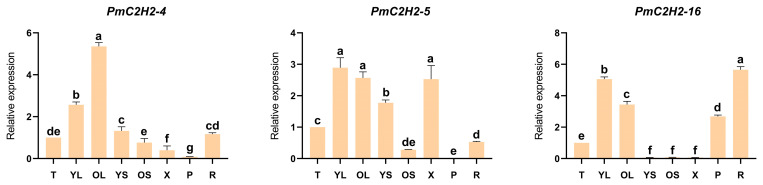
The relative expression levels of six Q-type *PmC2H2s* in different tissues: shoot apices (T), young needle leaf (YL), old needle leaf (OL), young stem (YS), old stem (OS), xylem (X), phloem (P), and root (R). The relative expression level in ‘T’ was set as “1”. The same lowercase letters between different columns indicate no significant difference. The highest column is marked with ‘a’, then ‘b’, and so on. Completely different lowercase letters between different columns indicate a significant difference, *p* < 0.05. More than one lowercase letter in the same column indicates no significant difference between the column and other columns that contain one of the lowercase letters.

**Figure 5 ijms-25-08361-f005:**
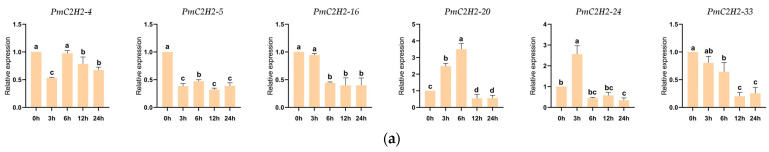
The figure above shows the expression levels of five *PmC2H2* genes under different abiotic treatments, namely (**a**) ABA, (**b**) drought, (**c**) ETH, (**d**) H_2_O_2_, (**e**) mechanical damage, (**f**) MeJA, (**g**) NaCl, (**h**) PEG, and (**i**) SA. The absence of any significant difference is indicated by the presence of identical lowercase letters across different columns. Conversely, the presence of completely distinct lowercase letters between different columns signifies a statistically significant difference (*p* < 0.05). In cases where multiple lowercase letters are present within the same column, it implies that there is no significant difference between that particular column and other columns containing any one of those lowercase letters. The relative expression at 0 h is normalized to “1”.

**Figure 6 ijms-25-08361-f006:**
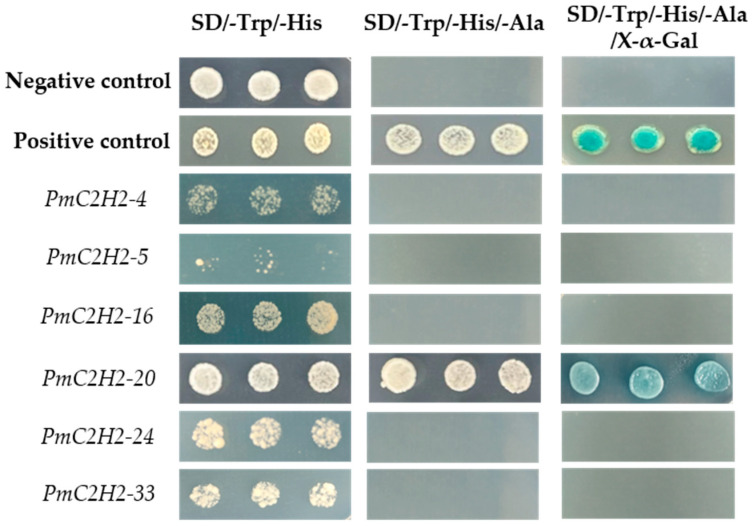
Transcriptional activation assay of six *PmC2H2* genes. Empty pGBKT7 vector was used as a negative control and pGBKT7-*PmC3H20* was used as a positive control.

**Table 1 ijms-25-08361-t001:** The characteristics of 57 C2H2 proteins identified and their subcellular localization prediction.

Gene ID	Aa	Mw (kDa)	pI	Subcellular Localization
*PmC2H2-1*	461	53.28	5.68	Nucleus
*PmC2H2-2*	688	76.54	8.58	Nucleus
*PmC2H2-3*	696	72.08	9.15	Nucleus
*PmC2H2-4*	559	61.93	5.47	Nucleus
*PmC2H2-5*	532	58.45513	6.23	Nucleus
*PmC2H2-6*	617	66.9	9.1	Nucleus
*PmC2H2-7*	395	44.67	6.59	Nucleus
*PmC2H2-8*	617	66.87	5.59	Nucleus
*PmC2H2-9*	522	57.31	5.49	Nucleus
*PmC2H2-10*	350	37.35	8.77	Nucleus
*PmC2H2-11*	598	65.21	8.94	Nucleus
*PmC2H2-12*	444	48.93	9.22	Nucleus
*PmC2H2-13*	385	43.03	5.93	Nucleus
*PmC2H2-14*	651	68.65	6.53	Nucleus
*PmC2H2-15*	725	76.17	9.19	Nucleus
*PmC2H2-16*	558	61.93	5.81	Nucleus
*PmC2H2-17*	411	46.68	5.43	Nucleus
*PmC2H2-18*	725	76.67	9.31	Nucleus
*PmC2H2-19*	239	27.41	8.01	Nucleus
*PmC2H2-20*	359	39.9	5.85	Nucleus
*PmC2H2-21*	400	45.44	5.47	Nucleus
*PmC2H2-22*	377	43.58	9.46	Nucleus
*PmC2H2-23*	424	46.51	6.65	Nucleus
*PmC2H2-24*	387	43.52	5.89	Nucleus
*PmC2H2-25*	351	37.69	4.82	Nucleus
*PmC2H2-26*	459	50.47	7.01	Nucleus
*PmC2H2-27*	434	47.89	8.84	Nucleus
*PmC2H2-28*	719	81.74	5.28	Nucleus
*PmC2H2-29*	282	30.88	8.93	Chloroplast
*PmC2H2-30*	298	32.77	8.06	Nucleus
*PmC2H2-31*	262	30.15	7.21	Nucleus
*PmC2H2-32*	414	45.99	8.84	Nucleus
*PmC2H2-33*	619	69.2	5.88	Nucleus
*PmC2H2-34*	415	46.88	6.68	Nucleus
*PmC2H2-35*	149	16.2	9.42	Nucleus
*PmC2H2-36*	156	17.88	9.2	Nucleus
*PmC2H2-37*	616	67.23	5.63	Nucleus
*PmC2H2-38*	206	22.94	8.53	Nucleus
*PmC2H2-39*	196	20.93	9.1	Nucleus
*PmC2H2-40*	202	22.12	9.51	Nucleus
*PmC2H2-41*	155	16.99	9.33	Nucleus
*PmC2H2-42*	326	34.651	7.74	Nucleus
*PmC2H2-43*	665	69.54	5.63	Nucleus
*PmC2H2-44*	181	19.99	9.71	Nucleus
*PmC2H2-45*	140	16.17	9.87	Nucleus
*PmC2H2-46*	298	32.7	8.41	Nucleus
*PmC2H2-47*	228	25.1	7.1	Nucleus
*PmC2H2-48*	521	59.88	6.36	Nucleus
*PmC2H2-49*	520	57.43	8.18	Nucleus
*PmC2H2-50*	219	24.31	8.79	Nucleus
*PmC2H2-51*	468	53.08	8.61	Nucleus
*PmC2H2-52*	367	40.87	8.21	Nucleus
*PmC2H2-53*	548	60.01	4.96	Nucleus
*PmC2H2-54*	566	61.77	5.5	Nucleus
*PmC2H2-55*	137	16.09	9.5	Nucleus
*PmC2H2-56*	205	23.6	6.38	Nucleus
*PmC2H2-57*	177	19.21	9.1	Nucleus

**Table 2 ijms-25-08361-t002:** Sequences of the 10 motifs of PmC2H2 ZFPs.

Motif	Length	Motif Consensus	Motif Logo
1	21	CEICGKGFSRPQNLQQHMRTH	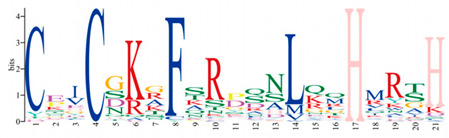
2	15	GCGKRFSVVSDLKRH	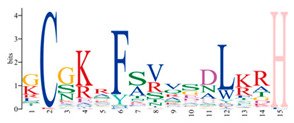
3	50	WKLRQRTTKEIRKRVYICPEPTCVHHDPSRALGDLTGIKKHFCRKHGEKK	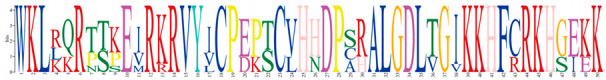
4	32	SKTCGTREYRCDCGTLFSRRDSFITHRAFCDA	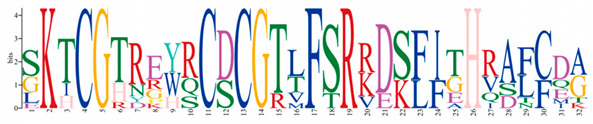
5	41	YSCPFEGCRRNKBHPKFKPLKSIRSLRNHYKRSHCPKMYTC	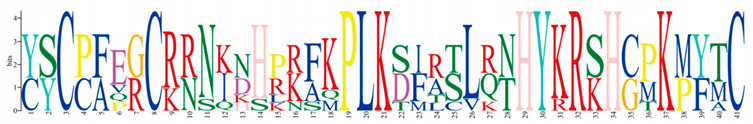
6	37	QQNTTIKRKRNLPGTPDPDAEVIALSPKTLMATNRFV	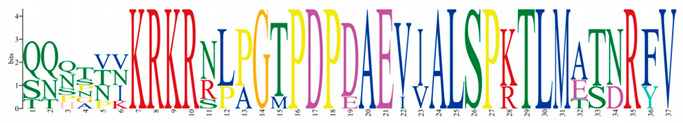
7	50	YSSASESVDLPNSRTLPRPSALVGGSMPPAPQSMMGQFSSKVSSSSQKKH	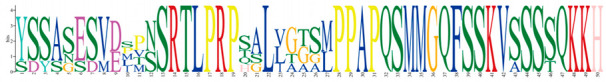
8	21	CSICGRSFTSKQALKGHIRVH	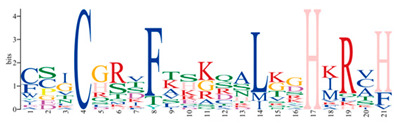
9	8	TGEKPFVC	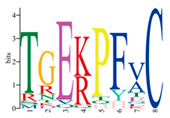
10	29	QQQQHTSTPQMSATALLQKAAQMGATASN	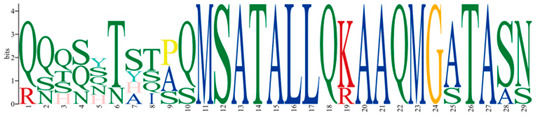

## Data Availability

The data presented in this study are available in Appendix A.

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
