# Peer review of "Transcriptomic Identification of Potential C2H2 Zinc Finger Protein Transcription Factors in Pinus massoniana in Response to Biotic and Abiotic Stresses"

_ijms, 2024, doi:10.3390/ijms25158361_

Round 1

Reviewer 1 Report

Comments and Suggestions for Authors

In the methodology section, there is a need for significant improvement.

1. Numerous software used to assess the bioinformatics analysis are not properly cited, such as Mega X, retools.

2. is there any specific reason for selecting the ML method in phylogeny? do the authors predict the most suitable method for this purpose? 

3. The analysis of cis-regulatory elements is missing, it is an important aspect, and I recommend including it.

4. The gene names in qRT-PCR assays should be italicized, as it is more about gene name, not protein

5. What are LD, MD, and SD? It will be inconvenient for new researchers to read this manuscript, so I suggest using their full name first, later on, they can use abbreviations.

6. line 369-370, "The mechanical damage treatment method was performed by cutting the upper half of the needle" What does that mean?

7. What about CK? how it was performed and enlist the methodology as well

8. The conclusion should be brief and in a single paragraph, i recommend revising it

9. Figure 1, 6 need further elaboration

10. is there any reason for highlighting the paragraph of subcellular localization bold?

Reviewer 2 Report

Comments and Suggestions for Authors

Dear Authors!

The manuscript and data presented are interesting and well presented. Some minor mistakes were found in the manuscript, as pointed below:

Row 130-136: the body text is with bold

Row 138-139: Figure 2 needs some more explanation (legend with the meaning of chloroplast, bright, merge etc.).

Row 238, Figure 6. Positive control is mispelled (posotive)

Rows 344, 347: Agrobacterium should be Italics

Row 372, 386: H2O2 and H2O (the numbers should be writen as subscript)

Round 2

Reviewer 1 Report

Comments and Suggestions for Authors

The authors have significantly modify the manuscript as per suggestion and i believe the article is ready for acceptance.